# Novel Splice Site Mutation in the *PROS1* Gene in a Polish Patient with Venous Thromboembolism: c.602-2delA, Splice Acceptor Site of Exon 7

**DOI:** 10.3390/medicina56090485

**Published:** 2020-09-22

**Authors:** Magdalena Mrożek, Ewa Wypasek, Martine Alhenc-Gelas, Daniel P. Potaczek, Anetta Undas

**Affiliations:** 1The John Paul II Hospital, 80 Prądnicka Street, 31-202 Krakow, Poland; magdalena.wojcik07@gmail.com (M.M.); ewa.wypasek@wp.pl (E.W.); mmundas@cyf-kr.edu.pl (A.U.); 2Faculty of Medicine and Health Sciences, Andrzej Frycz Modrzewski Krakow University, 30-705 Krakow, Poland; 3Laboratoire d’Hémostase, Service de Médecine Vasculaire, Centre Claude Bernard de Recherche sur les Maladies Vasculaires, Hôpital Broussais–AP-HP, 75116 Paris, France; martine.alhenc-gelas@aphp.fr; 4Société Serbio, 92230 Gennevilliers, France; 5Institute of Laboratory Medicine and Translational Inflammation Research Division & Core Facility for Single Cell Multiomics Members of the German Center for Lung Research (DZL) and Universities of Giessen and Marburg Lung Center (UGMLC), Philipps-University Marburg, 35043 Marburg, Germany; 6Institute of Cardiology, Jagiellonian University Medical College, 30-705 Krakow, Poland

**Keywords:** protein S, *PROS1* gene, splice site mutation

## Abstract

We identified a novel splice site mutation of the *PROS1* gene in a Polish family with protein S (PS) deficiency and explored the molecular pathogenesis of this previously undescribed variant. A novel mutation was detected in a 26-year-old woman with a history of venous thromboembolism (VTE) provoked by oral contraceptives. Her family history of VTE was positive. The sequence analysis of the *PROS1* gene was performed in the proband and the proband’s family. The proband and their asymptomatic father had lower free PS levels (45% and 50%, respectively) and PS activity (48% and 44%, respectively). Total PS levels were normal (65.6% and 62.4%, respectively). The sequence analysis of the *PROS1* gene revealed the presence of heterozygous deletion at the nucleotide position c.602-2 in intron 6, just upstream of exon 7, detected in the proband and her father. This variant alters the splice acceptor site of exon 7, and, according to the in silico prediction, it is highly likely to cause in-frame exon 7 skipping. We also presented follow-up data of two other Polish patients with PS deficiency associated with splice site mutations in *PROS1* gene.

## 1. Introduction

Protein S (PS) is a vitamin K-dependent glycoprotein that serves as a cofactor of activated protein C (PC) in the proteolysis of activated factor (F) V and FVIII [1]. PS was also identified as a cofactor of tissue factor pathway inhibitor (TFPI) which stimulates the inhibition of FXa by TFPI [2]. PS deficiency is an autosomal dominant disorder with a prevalence of less than 0.5% in the European population and 2% to 12% in patients with venous thromboembolism (VTE), which includes deep-vein thrombosis (DVT) and pulmonary embolism (PE) [3]. The annual incidence of VTE in PS deficient patients is estimated to be of about 0.8% [4].

PS is encoded by the *PROS1* gene which is approximately 80 kb long and consists of 15 exons and 14 introns [5]. PS deficiency is most frequently caused by missense/nonsense substitutions followed by splice site mutations and small/gross duplications, insertions or deletions. Until now there have been about 360 genetic mutations associated with PS deficiency (HGMD database, http://www.hgmd.org). In 2013 the first Polish PS deficient patient was reported [6], and in 2017, we published the first Polish cohort study involving 27 VTE patients with PS deficiency [7].

A splice site mutation alters nucleotides at splice consensus sequences, changing the patterns of RNA splicing [8]. To our knowledge, as few as 44 splice site mutations have been reported (HGMD database, http://www.hgmd.org) among PS-deficient subjects to date, including two *PROS1* gene mutations observed in Polish patients with VTE, i.e., c.1155+5G>A and c.965+4A>G [7]. The current report presents a novel splice site c.602-2delA mutation detected in a Polish family with PS deficiency, together with a long-term follow-up of two other cases with VTE associated with splice site mutations in the *PROS1* gene, treated in our Centre for Coagulation Disorders.

## 2. Materials and Methods

### 2.1. Patients

A 26-year-old Polish woman, after documenting the first VTE episode, was referred to our Centre for Coagulation Disorders to be tested for thrombophilia. The woman at the age of 22, while using oral contraceptives for about one year, experienced PE associated with DVT in the left lower extremity. Venous duplex ultrasound showed extensive proximal DVT involving the external and common iliac veins, with thrombus propagation to the inferior vena cava. Unfractionated heparin infusion was initiated due to intermediate-high risk PE and massive DVT; on admission, due to the patient’s condition, thrombolysis was considered within the first 48 h. The patient’s sPESI was 3 and cardiac troponin T was 59 ng/mL [9]. Then rivaroxaban at a dose of 15 mg bid was administered for 21 days. The rivaroxaban treatment at a dose of 20 mg daily was continued for 3 years, however, she experienced persistent clinically relevant bleeding, i.e., recurrent heavy menstrual bleeding (HMB) with the subsequent anemia (hemoglobin below 10 g/dL) requiring long-term iron supplementation. Based on the patient’s complaints and preferences, apixaban (5 mg twice daily) was administered instead of rivaroxaban. As quickly as after one month of treatment with apixaban and the first menstrual bleeding, HMB subsided. The follow-up of 12 months while treated with apixaban was unremarkable without VTE recurrences. Hemoglobin returned to normal.

Family history of VTE was positive. The proband’s grandmother on the paternal side experienced DVT at a young age. The proband’s parents and younger brother remain asymptomatic. During 12 months of follow-up, her family history was unchanged in relation to VTE episodes.

### 2.2. Laboratory Measurements

The proband was screened for thrombophilia including the assessment of the *F5* G1691A (FV Leiden, rs6025) and *F2* G20210A (rs1799963) polymorphisms, antithrombin (AT), protein C (PC) and factor VIII, along with markers of antiphospholipid syndrome. Immunoassays for free PS, total PS and PS activity were performed also in the proband’s parents and brother. Free PS levels were quantified using the immunoturbidimetric free protein S reagent (reference range, female: 60–114%, men: 67–139%; INNOVANCE Free Protein S Antigen; Siemens Healthcare Diagnostics, Erlangen, Germany), PS activity using the Protein S Activity Assay (reference range, 58–128%; Siemens Healthcare Diagnostics) and total PS using the Asserachrom Kit (reference range, 60–140%, unless otherwise indicated; Diagnostica Stago, Asnieres, France).

### 2.3. Genetic Testing

After obtaining informed written consent from the patient and patient’s parents, genomic DNA was extracted from whole blood samples. DNA from the patient’s brother was unavailable. All exons, exon–intron boundaries and the 700 bp promoter of the *PROS1* gene were subjected to PCR-based DNA sequencing using dideoxynucleotides at appropriate concentrations. In silico analysis was performed using Alamut Visual software v2.10 (Interactive Biosoftware, Rouen, France). Confirmatory analyses were conducted with Human Splicing Finder 3.1 program (http://www.umd.be/HSF/).

## 3. Results

Thrombophilia screening of the proband and her father did not show any abnormalities apart from low PS levels. Deficiencies in PC and AT as well as antiphospholipid syndrome were excluded. FV Leiden and *F2* gene G20210A mutation were absent. The free PS levels were 45% in the proband (patient 1, Table 1) and 50% in her father. The PS activity was 48% in the proband and 44% in her father. The total PS levels were normal for both subjects, 65.6% in the proband and 62.4% in her father. Type III PS deficiency was diagnosed. PS levels and activity were within the normal range in the patient’s mother and brother.

The sequence analysis of the PROS1 gene revealed the presence of heterozygous deletion at nucleotide position c.602-2 in intron 6, just upstream of exon 7 [NM_000313.3: c.602-2delA; Ch3 (GRCh38): g.93900931del], detected in the proband and her father. Bioinformatic data analysis performed using the Alamut Visual software v2.10 showed that the presence of this intronic mutation leads to a damaging alteration of the exon 7 splice acceptor site (scores: NNSPLICE = 100; MaxEnt = 100.0%; and HSF = 100.0%), most probably causing the in-frame exon 7 skipping. The same effect of the c.602-2 mutation was confirmed with the Human Splicing Finder 3.1 program.

## 4. Discussion

Here we report a novel *PROS1* gene c.602-2delA splice site mutation in a young woman with VTE with type III PS deficiency. According to the in silico prediction, this variant alters the splice acceptor site of exon 7, and, thus, it is very likely to cause in-frame exon 7 skipping (Figure 1).

Exon skipping has been reported as the most common alternative splicing event, which due to loss of functional domains/sites or shifting of the open reading frame may lead to synthesis of the shortened but still functional protein, despite the genetic mutation [10].

To date, there have been a few reports describing splice site mutation and exon skipping in the *PROS1* gene in patients with type I PS deficiency [11,12]. Menzenes et al. reported that a splice site mutation c.1871-14T>G in the *PROS1* gene was causative of type I PS deficiency in two families with VTE. This splicing variant probably results in a reduced synthesis of PS reflected by lower levels of free and total PS [11]. Mizukami et al. showed that the type I PS deficiency in a 25-year-old Japanese male patient, who developed PE associated with DVT, was caused by variant termed PS Sapporo 1 [12]. The heterozygous A-to-T change in the invariant AG dinucleotide of the acceptor splice site of intron f of the *PROS1* gene was identified. This splice site mutation impaired normal mRNA splicing, leading to exon skipping. The skipping in the mutant allele induced an in-frame deletion of 126-bp segments responsible for a deletion of 42 amino acids corresponding to the epidermal growth factor-like domain 3 (EGF3) of mature PS. Translation of this mutant transcript may result in a truncated protein that lacks the entire EGF3 of the PS molecule [12].

A novel splice site c.602-2delA mutation in the *PROS1* gene detected by us probably results in a reduced synthesis of PS reflected in lower levels of free PS and PS activity identified also in the proband’s father, who putatively is also at increased risk of VTE. Of note, a VTE episode occurred at a young age (22 years) in our patient, which is probably related to the effect of the *PROS1* c.602-2delA mutation on the regulation of blood clotting enhanced by oral contraceptives. It might be speculated that the patterns observed by Mizukami et al. were similar to ours, and additional factors, e.g., ethnicity related differences, affected total PS levels, which might explain why PS levels were just above the lower limit of the reference range in the present proband.

Regarding therapeutic issues, our patient was treated with rivaroxaban (20 mg/d), and this therapy was associated with HMB, which became unacceptable for the patient, and, therefore, apixaban therapy (2 × 5 mg/d) was initiated, leading to normal menses. In patients who are at increased risk of bleeding, the 10 mg of rivaroxaban is usually followed by 2.5 mg of apixaban twice daily. However, in those deficient in natural anticoagulants, which is associated with higher risk of recurrent VTE (A. Undas, unpublished data), we use an individual approach and apply the treatment with full-dose novel oral anticoagulants (NOACs). Currently, there is no specific guidance on the choice of anticoagulation for managing VTE in patients with inherited or acquired thrombophilia. Moreover, conflicting reports have been published regarding the efficacy of direct oral anticoagulants (DOACs) in preventing recurrent VTE in patients with protein C and S deficiency [13]. Therefore, our choice of the optimal anticoagulant strategy in PS-deficient patients is based on small observational studies and, in practice, follows the protocols developed in our center. In the current report, the anticoagulant therapy was also maintained by choice of the patient [14].

These observations are in line with previous studies indicating that HMB occurs relatively frequently in young women on rivaroxaban, and this side effect may increase the risk of recurrent VTE largely due to interrupted anticoagulant therapy [15]. Myers et al. suggested that a way to avoid HMB rivaroxaban could be by switching to apixaban, given the rate of HMB observed on apixaban of 9.3% [16]. Moreover, Wypasek et al. suggested that PS deficiency of <20% resulting from functional *PROS1* mutations might be associated with a weaker anticoagulant response to rivaroxaban therapy [17]. This observation based on two PS-deficient cases might provide additional argument supporting the use of a full-dose apixaban in young women with PS deficiency following VTE. Our case highlights challenges of long-term anticoagulation in women at a reproductive age and the need for individualized decision-making to balance the risk of recurrent VTE against the risk of bleeding, especially among thrombophilic patients.

In conclusion, we report here a novel splice site mutations in the *PROS1* gene in a Polish woman with type III PS deficiency and VTE. Our experience supports the use of apixaban 5 mg/d in PS-deficient VTE female patients as a preferred therapeutic option. Our data contribute to a better understanding of the genetic background of inherited PS deficiency in terms of co-segregation between the disease and the *PROS1* gene locus.

## Figures and Tables

**Figure 1 medicina-56-00485-f001:**
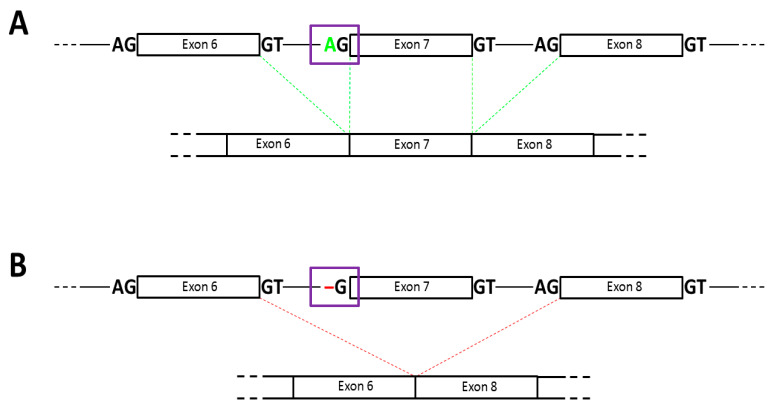
Putative functional consequences of the c.602-2delA mutation of the *PROS1* gene. Wildtype is associated with normal splicing (**Panel A**). The mutation damages the exon 7 splice acceptor site, most probably leading to the in-frame exon 7 skipping (**Panel B**). Black-outlined rectangles correspond to *PROS1* gene exons, while black horizontal lines connecting them to the introns. Violet-outlined rectangles indicate the wild-type (**Panel A**) or the mutated (**Panel B**) exon 7 splice acceptor site.

**Table 1 medicina-56-00485-t001:** Characteristics of the proband with protein S deficiency associated with a novel c.602-2delA (g.78160del) splice site mutation in the *PROS1* gene.

Sex/Age (years)	Free PS [60–114%]	PS Total [60–140%; 75–101%] ^†^	PS Activity [58–128%] *	Type of PS Deficiency	Clinical Manifestation	Age of First Thromboembolic Event	Family History of VTE	Duration of Follow-Up (months)	Treatment	Reccurent Thromboembolism
Female/26	45; 48	65.6; 77.2	48; 54	III	DVT + PE/oral contraception	22	Yes	12	Rivaroxaban 20 mg/d; apixaban 2 × 5 mg/d	No

DVT—deep vein thrombosis, PE—pulmonary embolism. * Repeated measurement of the sample. Reference ranges are given in square brackets. ^†^ Two reference ranges for two distinct reagent lots are given.

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
