# Peer review of "Novel Splice Site Mutation in the PROS1 Gene in a Polish Patient with Venous Thromboembolism: c.602-2delA, Splice Acceptor Site of Exon 7"

_medicina, 2020, doi:10.3390/medicina56090485_

Round 1

Reviewer 1 Report

In a manuscript entitled “Novel splice-site mutation in the PROS1 gene in a Polish patient with venous thromboembolism: c.602-2delA, splice acceptor site of exon 7”, Magdalena Mrożek et al reported a novel PROS1 variant, c.602-2delA, resulting in splicing abnormality. The patient carrying the mutation showed a severe thrombophilic phenotype.

Major comments

  1. Proband’s father has a same PROS1 However, in the family history of VTE, proband’s maternal, not paternal, grandmother has the experience of VTE event. I can not understand the relationship of the PROS1 variant and family pedigree. Is this a coincidence? Did authors investigate a maternal grandmother’s genetic variation which associated with thrombophilia?
  2. In silico analysis using Alamut Visual software v2.10 predicted that c.602-2delA resulted in an exon 7 skipping due to a disruption of intron 6 splice acceptor site. I recommend that authors should re-check the simulation result using by other splicing simulation software (e.g., Splice Site Prediction.)
  3. Author should indicate the mutation type at a protein level. Is this exon 7 skipping an in-frame mutation or not? If this is the in-frame mutation, authors should show the predicted mutant PS structure. Moreover, if the mutation causes a frameshift, c.602-2delA could be a null mutation. In the manuscript, authors mentioned that the patient was diagnosed as Type III PS deficiency. However, if the frameshift mutation, c.602-2delA must be Type I PS deficiency. (actually, the patient’s total PS antigen level seems to be low)
  4. In Table 1, PS activity was not shown.
  5. In Table 1, authors must present a normal range of your hospital for each clinical test.
  6. In Table 1 and discussion section, patient ID 2 and 3 were not necessary. These follow-up data do not match a subject of the manuscript. The title of the manuscript is only focusing on the PROS1 602-2delA.

Minor comments

  1. Page 3 Line 114, Figure S1 is Figure 1.

Author Response

Reviewer 1

Major comments

1. Proband’s father has a same PROS1 However, in the family history of VTE, proband’s maternal, not paternal, grandmother has the experience of VTE event. I can not understand the relationship of the PROS1 variant and family pedigree. Is this a coincidence? Did authors investigate a maternal grandmother’s genetic variation which associated with thrombophilia?

Indeed, we did made a spelling mistake, and the incident was observed in the proband’s paternal grandmother, not maternal. The error has been corrected.

2. In silico analysis using Alamut Visual software v2.10 predicted that c.602-2delA resulted in an exon 7 skipping due to a disruption of intron 6 splice acceptor site. I recommend that authors should re-check the simulation result using by other splicing simulation software (e.g., Splice Site Prediction.)

Additional analyses performed using Human Splicing Finder 3.1 confirmed the effects suggested by Alamut Visual software v2.10 software. This information has been added, as suggested by the Reviewer.

3. Author should indicate the mutation type at a protein level. Is this exon 7 skipping an in-frame mutation or not? If this is the in-frame mutation, authors should show the predicted mutant PS structure. Moreover, if the mutation causes a frameshift, c.602-2delA could be a null mutation. In the manuscript, authors mentioned that the patient was diagnosed as Type III PS deficiency. However, if the frameshift mutation, c.602-2delA must be Type I PS deficiency. (actually, the patient’s total PS antigen level seems to be low)

At a protein level, skipping of the 126 bp of exon 7 (whole exon 7) leads to an in-frame deletion, resulting in a truncated PS without the entire epidermal growth factor-like 3 (EGF3) domain. In more detail, this truncated protein lacks 42 internal amino acids, including six cysteines involved in the formation of three disulfide bridges and a consensus sequence associated with high-affinity Ca2+ binding, both of which play important roles in proper folding during assembly of the ternary structure of the EGF3 domain. Thus, substantial functional consequences of our novel mutation are to be expected. As suggested by the Reviewer, the mutation type has been indicated at a protein level and the consequences of the in-frame exon 7 skipping have been shortly described.

To verify the type of deficiency, we repeated the measurements of total and free PS and PS activity in the proband (for the results, please, refer to a new version of Table 1). Total PS levels were close the lower limit of the reference range and were similar to those obtained previously (77.2% in proband and 92.2% [75 -101%] in her father). All the values from 2 separate measurements confirmed that a novel PROS1 mutation, i.e. skipping of the 126 bp of exon 7, is associated with type III PS deficiency.

Interestingly, a very similar mutation has been described by Mizukami et al. (Am. J. Hematol. 2006;81:787–797, current ref 11); the difference was that in their case it was c.602-2A>T. The consequences for the splicing and protein synthesis were, however, comparable, though type I PS deficiency was detected by Mizukami et al. On the other hand, the effects observed by Mizukami et al. were the strongest for free PS levels and PS activity and the weakest for total PS levels, which is consistent with the current results. One might thus speculate that the patterns observed by them were similar to ours and additional factors, e.g. ethnicity related differences, affected total PS levels and they might explain why in the present proband PS levels were just above the lower limit of the reference range. The appropriate comments have been provided in the revised manuscript.

4. In Table 1, PS activity was not shown.

In Table 1, PS activity of the index patient has been added.

5. In Table 1, authors must present a normal range of your hospital for each clinical test.

According to the Reviewer’s request, in Table 1, a normal (reference) range used in our hospital for each clinical test has been added.

6. In Table 1 and discussion section, patient ID 2 and 3 were not necessary. These follow-up data do not match a subject of the manuscript. The title of the manuscript is only focusing on the PROS1 602-2delA.

As requested by the Reviewer, patients ID 2 and 3 have been removed from Table 1. Likewise, the corresponding plain text has been deleted from the discussion section.

Minor comments

1. Page 3 Line 114, Figure S1 is Figure 1.

The name of the figure has been corrected.

Reviewer 2 Report

interesting paper, due to the description of the ps mutation gene. 

on line 64 the authors describe as raccomended these Unfractionated heparin followed by rivaroxaban, this is not correct, please change.

on line 65 the authors describe the use of rivaroxaban 20 mg for 3 years for first episode of first episode of provoked Vte. It's not as for guidelines.

on line 69 the authors change from full dose rivaroxaban to full dose Apixaban why don't the choose Half dose of Apixaban (2,5 mg did) as for the ext study.

on line 143, in the discussion the authors suggest the same thing (change to rivaroxaban to Apixaban same dose) but we don't have any evidence that this is correct, only a correspondence regarding an observation. however, again, why don't the authors decide to use the half dose please describe

Author Response

Reviewer 2

Comments and Suggestions for Authors interesting paper, due to the description of the ps mutation gene. 

on line 64 the authors describe as recommended these Unfractionated heparin followed by rivaroxaban, this is not correct, please change.

For intermediate-high or high PE patients heparin therapy is often administered with close surveillance to facilitate thrombolysis or invasive therapy if the patient condition worsens. After stabilization and reduced arterial pulmonary pressure, decision to start oral anticoagulation is made. We have followed this strategy in our proband.

on line 65 the authors describe the use of rivaroxaban 20 mg for 3 years for first episode of first episode of provoked VTE. It's not as for guidelines

Decision to continue anticoagulation in the proband was related to the extensive DVT and PE associated with PS deficiency, which was diagnosed within the first year since the index event. This issue has been clarified in the revised manuscript.

on line 69 the authors change from full dose rivaroxaban to full dose Apixaban why don't the choose Half dose of Apixaban (2,5 mg did) as for the ext study.

on line 143, in the discussion the authors suggest the same thing (change to rivaroxaban to Apixaban same dose) but we don't have any evidence that this is correct, only a correspondence regarding an observation. however, again, why don't the authors decide to use the half dose please describe

The decision to use full-dose apixaban was driven by the diagnosis of PS deficiency. Our experience, including the published case reports (Wypasek E et al., Thrombosis Research. 2014; 134; 199-201), indicates that patients with this thrombophilia should be treated with a full-dose NOACs unless there are contraindications or unacceptable bleeding tendency. Despite the fact that some experts consider PS deficiency as a non-severe thrombophilia, in our center massive thrombosis with the post-thrombotic syndrome together with PE in patients deficient with PS, as well as PC or AT, with a positive family history is treated in most cases with full-dose NOACs. Based on our experience, use of half-dose NOACs in patients deficient in natural anticoagulants is associated with higher risk of recurrent VTE (A. Undas, unpublished data). We are aware of the fact that the choice of the optimal anticoagulant strategy in PS deficient patients is largely based on case reports and small observational studies and, in practice, follows the protocols developed in the given center. To highlight the controversy around this issue, we have added an additional paragraph to the revised Discussion section.

Round 2

Reviewer 1 Report

Authors adequately responded to my comments.

Thanks.

Author Response

Thank you very much.

Reviewer 2 Report

i appreciate the changes but to describe an intermediate-hight risk pe you need to describe also spesi score and troponin (as for esc guideline 2020 witch must be added in the references).

on line 162 you can also add that the anticoagulant therapy was also maintained by choice of the patient, as for accp guidelines witch must be added in the reference 

Author Response

I appreciate the changes but to describe an intermediate-high risk PE you need to describe also spesi score and troponin (as for esc guideline 2020 witch must be added in the references).

The sPESI score and troponin together with the reference have been added as suggested.

on line 162 you can also add that the anticoagulant therapy was also maintained by choice of the patient, as for ACCP guidelines witch must be added in the reference 

The information about anticoagulation therapy and reference have been added as suggested.